# Sugar Substitute Stevia Inhibits Biofilm Formation, Exopolysaccharide Production, and Downregulates the Expression of Streptococcal Genes Involved in Exopolysaccharide Synthesis

**DOI:** 10.3390/dj11120267

**Published:** 2023-11-23

**Authors:** Sara AlKanderi, Monerah AlFreeh, Radhika G. Bhardwaj, Maribasappa Karched

**Affiliations:** Oral Microbiology Research Laboratory, Department of Bioclinical Sciences, College of Dentistry, Kuwait University, Safat 13110, Kuwait; sara.alkanderi@hscd.ku.edu.kw (S.A.); monerah.alfreeh@hscd.ku.edu.kw (M.A.); guleri.guleri@ku.edu.kw (R.G.B.)

**Keywords:** artificial sweeteners, stevia, *Streptococcus mutans*, *Streptococcus gordonii*, sucrose fermentation, dental caries, biofilm, glucosyltransferases, glucan-binding proteins

## Abstract

Background: Acid production by sucrose fermentation disturbs the balance in dental plaque by lowering the oral pH. As a consequence of the profound effect of sucrose on caries initiation and progression, many studies have been directed towards finding non-cariogenic artificial sweeteners that can be used as a substitute to sucrose. Existing literature shows that dietary sucrose upregulates the expression of biofilm associated genes involved in exopolysaccharide (EPS) production. Objective: In this study, we aimed to investigate the effect of the sugar substitute stevia on biofilm formation, EPS secretion, and streptococcal genes encoding glucan-binding proteins (Gbps) and glucosyltransferases (Gtfs), which are essential for the synthesis of EPS. Materials and Methods: *Streptococcus mutans* and *Streptococcus gordonii* were grown as biofilm cultures with or without stevia and sucrose. Biomass was quantified for biofilm and EPS production by crystal violet staining and the phenol–sulfuric acid method, respectively. Expression of *gtfB* and *gbpB* genes was studied by RT-PCR. Results: The quantities of biofilm were significantly lower when grown in the presence of stevia compared to sucrose in both species (*p* < 0.05). The proportion of EPS in the biofilm pellet decreased with increasing concentrations of stevia in both species but remained nearly unchanged with sucrose with respect to the control. In both streptococcal species, exposure of stevia decreased the expression of *gtfB* and *gbpB* genes compared to sucrose (*p* < 0.05). In comparison to the untreated control, the expression was decreased in the presence of stevia in both species, while it increased 2.5- to 4-fold in *S. mutans* and 1.5- to 2.5-fold in *S. gordonii* in the presence of sucrose. Conclusion: The ability of stevia to inhibit biofilm formation, reduce EPS production, and downregulate the expression of *gtfB* and *gbpB* genes in *S. mutans* and *S. gordonii* may have potential therapeutic applications in controlling dental plaques and caries.

## 1. Introduction

Tooth decay or dental caries is a multifactorial non-communicable chronic disease which is caused by the disturbance in the balance of demineralization and remineralization processes in the oral cavity [1,2,3,4,5]. Cariogenic bacteria in dental plaque produce acids by fermentation of dietary carbohydrates (such as sucrose which is the most cariogenic sugar) that can demineralize tooth enamel, leading to the formation of cavities [6,7]. The acid production by sucrose fermentation disturbs the balance in the plaque by lowering the oral pH and according to the ecological plaque hypothesis this low pH results in an increase in cariogenic bacteria in the plaque [8]. Long-term dietary sugar consumption and in situ experimental studies have also supported this hypothesis [9,10].

In addition to its ability to be fermented by oral bacteria, sucrose acts as a substrate for the synthesis of extracellular polysaccharide/exopolysaccharide (EPS) and intracellular polysaccharide by the cariogenic bacteria in the dental plaques (also known as dental biofilms). Among cariogenic bacteria, *Streptococcus mutans* has been firmly established as the predominant cariogenic pathogen in dental caries [11,12]. It has the ability to produce large amounts of organic acids by the fermentation of dietary carbohydrates (most commonly, sucrose and fructose) and to survive in such a persistent acidic environment. In addition, *S. mutans* produces glucosyltransferases (Gtfs) and multiple glucan-binding proteins (Gbps) that contribute to its remarkable ability to form cariogenic biofilms [13,14,15,16]. The sucrose-dependent adherence of *S. mutans* is mediated by metabolizing sucrose to glucans by Gtfs enzymes and then binding to these glucans through Gbps, initiating the synthesis of the EPS matrix, which is the major virulence determinant of cariogenic biofilm [17,18,19]. Existing literature also shows that dietary sucrose upregulates the expression of biofilm associated genes, e.g., *gtfB*, *gbpA*, *gbpB*, *gbpC*, *gbpD* involved in EPS production [19,20,21]. 

The EPS matrix in biofilms serves as a foundation for bacterial colonization, multi-species biofilm formation, and plays a role in pathogenesis [22,23], hence, playing a significant part in the development of dental caries and periodontal diseases such as gingivitis and periodontitis [18,24]. It also acts as a physical barrier, preventing antimicrobial agents from reaching the bacteria [25] and contributes to their persistence and recurrence even after professional dental cleanings. Therefore, understanding the factors influencing biofilm formation and developing strategies to disrupt it are crucial for maintaining oral health.

Since the significant relationship between dental caries and dietary sucrose consumption was established, several preventative measures have been tested to help reduce the incidence of caries including replacing sucrose with non-cariogenic artificial sweeteners such as, stevia [26,27]. Stevia is a non-caloric sweetener originally derived from a plant called *Stevia rebaudiana*. It has gained increasing popularity as a sugar substitute due to its natural origin, zero-calorie content, and high sweetness. It is around 50–350 times sweeter than sucrose when given in the same concentration. It can help reduce the total caloric intake and treat or prevent metabolic disorders. The rising rates of obesity, diabetes, and related comorbidities have led to global public policies advocating for reduced sugar consumption, further driving the interest in such low- and no-calorie sweeteners [27]. Its safety has also been affirmed by the European Food Safety Authority [28] and research has shown its potential health benefits, including its non-cariogenic nature and inhibitory effects on biofilm formation [27,29].

In this study, we hypothesized that stevia, a non-caloric sugar substitute, might have a substantial effect on both the biofilm mass and the expression of genes involved in biofilm matrix production. Thus, we aimed to investigate the effect of the sugar substitute stevia on biofilm formation, EPS production, and streptococcal genes encoding glucan-binding proteins and glucosyltransferases, which are essential for the synthesis of exopolysaccharides. 

## 2. Methods

### 2.1. Bacterial Strains and Culture Conditions

*S. mutans* CCUG 11877 and *Streptococcus gordonii* CCUG 33482 (CCUG, Gothenburg University, Gothenburg, Sweden) were cultured on brucella blood agar plates and incubated at 37 °C with 5% CO_2_ in air for two days. To ensure the purity of the cultures prior to each experiment, the plates were carefully examined under a stereo microscope. 

### 2.2. Biofilm Culture 

Biofilm cultures were prepared following the protocol described previously [30]. In brief, colonies obtained from the brucella blood agar cultures were suspended in brucella broth and the cell suspensions were adjusted to an optical density (OD) of 1 at the wavelength 600 nm (OD_600_ = 1). A 100 μL volume of suspension from each strain was added into separate wells of a 24-well cell culture plate containing 900 μL of brucella broth. Both stevia and sucrose were added to the culture medium at different concentrations (0, 1, 5, 10, and 25 mg/mL) in four separate sets. A well with broth only, but no biofilm, was used as a negative control.

### 2.3. Quantification of Biofilms

After two days of incubation of biofilm cultures, crystal violet staining (0.1% aqueous) was performed on the first set to quantify the biomass. For this, first, the supernatant broth from the wells was removed and discarded. The wells were then subjected to two washes with 1 mL of sterile phosphate-buffered saline (PBS) to remove unbound and loosely-bound cells. Following this, 1 mL of methanol was added to each well, and the plate was left undisturbed in a fume hood at room temperature for 15 min fixation period. The methanol was subsequently removed, and the 24-well plate was air-dried at room temperature for a duration of 45 min. Once dried, each well was stained with 1 mL of 0.1% aqueous crystal violet solution and incubated for 20 min at room temperature. Then, the stain was aspirated, and the 24-well plate was subjected to a thorough washing under running tap water a minimum of seven times, ensuring the complete removal of any excess stain. To eliminate any residual water in the wells, the plate was gently tapped onto dry tissue paper, and then left open in a fume hood for an additional 5 min at room temperature to facilitate complete drying. After this, photographs of crystal violet-stained biomass in the wells were captured. Further, for destaining, 500 µL of 33% acetic acid was added to each well containing the stained biomass and agitated by keeping the plate on a shaker for 5 min for effective destaining. Subsequently, 300 µL of the destained solution from each well of the 24-well plate was carefully transferred in triplicate to the wells of a 96-well plate, with each well containing 100 µL. Quantification was achieved by measuring the optical density at a wavelength of 590 nm using a spectrophotometer. This optical density reading provided a quantitative assessment of biofilm formation. 

### 2.4. Acid Production from Biofilms

To assess the pH levels, the biofilms (from the second set of biofilm cultures) were scraped off using sterile scrapers and transferred into a 15 mL tube, followed by vigorous vortexing for 1 min. Subsequently, pH readings were recorded by immersing a pH electrode into the biofilm suspension.

### 2.5. Exopolysaccharide (EPS) Quantification

From the third set of biofilm cultures, the EPS was quantified from the biofilms and the biofilm-supernatants following a previously published method with some modifications [31]. The crystal violet quantification results clearly indicated a notably greater biofilm biomass when sucrose was present. Recognizing that this could directly impact extracellular polymeric substance (EPS) levels, we collected an equal amount of biomass from both biofilms treated with sucrose and stevia for EPS quantification. The biofilms were scraped off from the 24-well plate wells, transferred to sterile microcentrifuge tubes, and centrifuged at 5000× *g* for 5 min. The pellets (50 mg from each well) were resuspended with 0.1 M NaOH. After centrifuging as above, the supernatant was passed through 0.22 µm syringe-driven filters. The filtered supernatant was subsequently subjected to precipitation by adding three volumes of ice-cold 95% ethanol and allowing it to incubate overnight at 4 °C. To the resultant EPS solution, one part of ice-cold 5% phenol together with five parts of concentrated sulfuric acid were added. The mixture was left to incubate at room temperature for 10 min until a red color developed. Then, the absorbance of the mixture was measured at 490 nm. To assess the extent of EPS production from the treated samples, the absorbance reading from the untreated control was considered as 100%. 

### 2.6. RNA Extraction

After two days of incubation, the broth supernatants from the fourth set of biofilm cultures were carefully withdrawn, and the biofilms were scraped off and collected into sterile microcentrifuge tubes. Subsequently, RNAlater (approximately 10 volumes corresponding to the biofilm weight) was immediately added to these tubes, and the samples were preserved at −20 °C until RNA purification. RNA from the samples was extracted by using RNeasy^®^ Maxi Kit (Qiagen GmbH, Hilden, Germany) following the manufacturer’s guidelines. The concentration of the RNA was determined using a Nanodrop, and its purity was assessed by the A_260_/A_280_ ratio. The RNA, eluted in sterile nuclease-free water, was preserved at −20 °C until used. 

### 2.7. Reverse Transcription Real-Time PCR

Purified RNA stored at −20 °C underwent cDNA synthesis through the application of a High-Capacity cDNA Reverse Transcription Kit (ABI Systems, Mississauga, ON, Canada), following the manufacturer’s recommended procedures. To normalize the expression levels of the genes *gtfB* and *gbpB,* the 16S rRNA gene was employed as an internal reference. The primer sequences used were selected from previously published literature [32,33]. The PCR reactions were conducted using an ABI 7000 Fast Real-Time PCR machine (Applied biosystems, San Francisco, CA, USA). The composition of the reaction mixture used was as follows: 4 µL 5× Hot FirePol qPCR Supermix (Evagreen; Solis BioDyne, Tartu, Estonia), 0.5 µL each of the forward and reverse primers (0.2 µM), 14 µL H_2_O, and 5 µL of DNA template. The temperature profile employed in the PCR protocol consisted of an initial denaturation at 95 °C for 10 min, followed by 40 cycles of denaturation at 95 °C for 15 s, annealing at 50–60 °C for 1 min, and a final extension at 60 °C for 1 min. The fluorescent signal data were acquired during the elongation step. The data were analyzed using SDS software 1.4.0v. The evaluation of target gene expression employed the comparative ∆∆Ct technique. The target quantity, normalized against the endogenous (16S rRNA gene) and in relation to the calibrator (untreated), was presented graphically using 2^−∆∆Ct^ to represent the fold change in expression. Two separate experiments were conducted for both CFU counts and gene expression. In each experiment, every sample was duplicated.

### 2.8. Statistics

All experimental procedures were independently replicated a minimum of two times to ensure robustness and reliability. Each experiment incorporated appropriate biological and technical replicates. Statistical comparisons between groups were carried out using the Mann–Whitney U test. Gene expression data were subjected to analysis through one-way analysis of variance (ANOVA) with a subsequent Tukey post-hoc test. A *p*-value of <0.05 was considered statistically significant. The statistical software SPSS vs28 (IBM, Chicago, IL, USA)for Windows was utilized for all analyses. 

## 3. Results

Both *S. mutans* and *S. gordonii* formed thick biofilms as revealed by crystal violet staining. The quantities of biofilm were significantly lower when grown in the presence of stevia compared to sucrose in both species (*p* < 0.05) (Figure 1). The pH of *S. mutans* biofilm dropped from 6.5 to 4.0 in the presence of sucrose, while in the presence of stevia, it was decreased from 6.5 to 5.2. On the other hand, *S. gordonii* biofilm pH was reduced from 6.6 to 4.8 in the presence of sucrose, while with stevia the pH remained unchanged at 6.5 (Appendix A). 

The EPS content of both *S. mutans* and *S. gordonii* biofilms showed a significant (*p* < 0.05) decrease across all concentrations of stevia, as compared to the negative control without stevia (Figure 2). However, this reduction was observed exclusively in the biofilm pellets and not in the biofilm supernatants. On the other hand, the biofilms of both species when grown in different concentrations of sucrose did not show a decrease in EPS content (Figure 2). Interestingly, in the case of S. mutans supernatant at a sucrose concentration of 5 mg/mL, exopolysaccharide production appeared to increase significantly by 100% compared to a sucrose concentration of 0 g/mL (*p* < 0.05), followed by a steep decline at the next concentrations of 10 and 25 mg/mL (Appendix A). 

In both *S. mutans* and *S. gordonii*, the expression of *gtfB* and *gbpB* genes exhibited a significant decrease at all tested concentrations of stevia when compared to that of sucrose (*p* < 0.05) (Figure 3). In *S. mutans*, when compared to untreated control, the *gtfB* expression decreased from 0.85-fold (at 1 mg/mL) to 0.36-fold (at 25 mg/mL) in the presence of stevia. Similarly, the *gbpB* expression decreased from 0.52-fold to 0.32-fold. The expression of these genes in *S. gordonii* also decreased with increasing stevia concentration. Conversely, when exposed to varying concentrations of sucrose, the expression of *gbpB* gene increased in both *S. mutans* and *S. gordonii* several fold higher than the untreated control. The expression of *gtfB* gene, however, remained relatively unchanged in *S. gordonii*, while it increased by approximately 2.5- to 4-fold in *S. mutans* (Figure 3).

## 4. Discussion

Biofilms are well-known contributors to the development of dental caries. Recently, stevia, a natural non-carbohydrate sweetener was investigated for its potential to prevent dental caries by disrupting biofilms, increasing pH levels, inhibiting the production of EPS, and regulating biofilm-related genes [29,34,35,36].

In the present study, to explore the potential impact of stevia on ardent dental colonizers, i.e., *S. mutans* and *S. gordonii,* we evaluated the effect of different concentrations of stevia on their biofilm formation and EPS production. We also investigated the effect of stevia on the expression of streptococcal genes encoding glucan-binding proteins (*gbpB* gene) and glucosyltransferases (*gtfB* gene), which are essential for the synthesis of extracellular polysaccharides. The results from crystal violet staining-based quantification revealed that stevia led to reduced biofilm formation compared to sucrose in both streptococcal species (*p* < 0.05). The EPS production was also found to be decreasing in a biofilm pellet of both species as the concentration of stevia increased (*p* < 0.05), whereas it remained relatively unchanged in the presence of sucrose compared to the untreated control. In both streptococcal species, exposure of stevia decreased the expression of *gtfB* and *gbpB* genes compared to sucrose (*p* < 0.05). Furthermore, in comparison to the untreated control, the expression was decreased in the presence of stevia in both species, while it increased 2.5- to 4-fold in *S. mutans* and 1.5- to 2.5-fold in *S. gordonii* in the presence of sucrose.

The potential effects of stevia on biofilm formation were investigated in several studies. It was found to inhibit the growth and biofilm formation of various bacteria, including *S. mutans, Borrelia burgdorferi, Lactobacillus,* and coagulase-negative staphylococci [29,37,38]. Furthermore, a study by Guo et al. demonstrated the inhibitory effects of stevioside on a dual-species biofilm model of *S. mutans* and *Candida albicans* [39], highlighting its potential against fungal species. In addition to inhibiting biofilm formation, stevia has also been shown to reduce EPS production [26,29], suggesting its ability to interfere with the structural integrity and protective properties of biofilms, making them more susceptible to disruption and removal. The mechanisms by which stevia exerts its effects on biofilm formation are not yet fully understood. However, it has been supposed to be due to the presence of bioactive compounds in it. Phenols have been identified as the predominant bioactive constituents (phytochemicals) followed by phytosterols, saponins, tannins, glycosides, and flavonoids [40]. These phytochemicals exhibit antimicrobial activities through diverse mechanisms. Tannins, for instance, form irreversible complexes with proline-rich proteins, inhibiting bacterial cell wall synthesis; saponins can induce protein and enzyme leakage from bacterial cells; and glycosides have been linked to inhibit the growth of cariogenic bacteria [41,42].

The finding of our study reveals that stevia decreases biofilm formation and EPS production in *S. mutans* and *S. gordonii* as compared to sucrose. This is consistent with several previous studies [26,29,34]. An interesting observation in our study was that at 5 mg/mL sucrose, *S. mutans* EPS production increased by about 100% as compared to 0 mg/mL sucrose, followed by a sharp decrease at higher sucrose concentrations. This is suggestive of a plateau stage leading to reduction in EPS production at sucrose concentrations higher than 5 mg/mL. Also, the pH of both *S. mutans* and *S. gordonii* biofilms in the current study was found to be reduced in the presence of sucrose, while with stevia it remained nearly unchanged. These findings are in line with the findings by Siraj et al. [43] who showed that rinsing a plaque with 10% sucrose solution led to a decrease in plaque pH when compared with rinsing with 0.2% aqueous solution of stevia leaf extract and a commercially available stevia product. Similar observations were made by Brambilla et al., with *Stevia rebaudiana* extracts on *S. mutans* biofilm formation and plaque pH, and the study concluded that *S. rebaudiana* extracts do not promote acidogenic metabolism from supragingival plaque bacteria [26].

A study by Das et al. where sixty *S. sobrinus*-colonized albino Sprague–Dawley rats were fed with 30% sucrose, 0.5% stevioside, and 0.5% rebaudioside A for 5 weeks showed that neither stevioside nor rebaudioside A were cariogenic [44]. Another study by Giacaman et al. assessed the cariogenic potential of several commercial sweeteners including stevia, sucralose, saccharin, aspartame, or fructose using an experimental biofilm caries model [45]. The results showed that stevia reduced biofilm formation as compared to sucrose, although complete eradication was not achieved. Similar findings were observed in the present study also where complete elimination of biofilm growth was not possible, but a significant reduction in the biofilm was noticed. It is pertinent to highlight a study conducted by Augustinho et al., which observed that some commercial formulations of stevia and aspartame exhibited cariogenic potential similar to that of sucrose, possibly due to the presence of lactose as an impurity in them. Conversely, the study revealed that the pure forms of these artificial sweeteners did not demonstrate cariogenic properties [46]. This suggests that the effectiveness of stevia may depend on its concentration (or purity) in addition to various other factors, such as the type of bacterial species and strains, quorum sensing, and signaling pathways that regulate biofilm formation and EPS production, as well as the presence of other dietary carbohydrates such as sucrose. A recent study demonstrated that the inhibitory effects of stevioside were significantly reduced on supplementation with sucrose [39].

The antimicrobial properties of stevia inhibiting biofilm formation by cariogenic bacteria has shown potential therapeutic applications in controlling dental biofilm and caries [47]. It has also been evaluated for its potential use in dental care products such as toothpaste and mouthwashes. A study by Usha et al. concluded that the effect of 0.5% aqueous extracts of stevia leaves is comparable to 0.12% chlorhexidine mouthwash in reducing the number of cariogenic bacteria (*S. mutans* and *Lactobacilli*) and improved the pH of saliva in high-risk caries patients [48].

The analysis of streptococcal genes involved in EPS synthesis and their regulation has provided insights into the mechanisms underlying biofilm formation and pathogenicity. In *S. mutans*, Gtfs and Gbps have been found to play a crucial role in the synthesis of EPS. A study by Sun et al. observed a significant upregulation of *gtfB* and *gbp* genes in *S. mutans* on exposure to high-fructose corn syrup and sucrose [20]. Furthermore, studies show that a deletion or mutation in *gtfB* gene of *S. mutans* reduced its biofilm formation and cariogenic abilities [49,50]. Additionally, *rnc* (encoding for ribonuclease III) gene has been found to regulate EPS synthesis in *S. mutans* through a genetic network involving formation-related and disintegration-related genes [51]. The results of the present study demonstrated the decreased expression of *gtfB* and *gbpB* genes with increased concentrations of stevia, suggesting the regulation of these EPS synthetizing genes and thereby influencing biofilm formation and EPS production.

Apart from the genes involved in EPS synthesis, the virulence and the cariogenic potential of *S. mutans* are also influenced by its acidogenic and aciduric properties; e.g., lactate dehydrogenase (ldh), encoded by *ldh* gene, stands out as the most significant enzyme responsible for acid production by *S. mutans*, as substantiated by previous studies [52], while the membrane-bound F-ATPase (H+-translocating ATPase) enzyme, encoded by *atpF* gene, plays a central role in conferring acid tolerance to it [53]. It is to be noted that the cariogenic potential of *S. mutans* does not arise from individual gene responses. Instead, it arises from a meticulously coordinated series of regulatory mechanisms [54]. Specifically, two-component regulatory systems, the VicRK signal transduction system, and the LiaSR regulatory system have been identified in regulating the expression of *gtfB*, *gtfC*, *gftD*, *gbpB*, and *ftf* (encoding for fructosyltransferase) genes in *S. mutans* and affecting the processes of adhesion, EPS production, biofilm formation, and genetic competence development in it [55,56]. Meanwhile, the modulation of transcription factors involved in gene regulation is one potential mechanism by which stevia may influence the expression of streptococcal genes. The transcription factors control the gene expression by binding to specific DNA sequences and activating or repressing target genes. In stevia, it has been reported that transcription factors such as WRKY, MYB, bHLH, and, NAC may participate in the regulation of secondary metabolites, including steviol glycosides [57].

This study has certain limitations. The oral microecology is a complex system with interactions between different microorganisms. Our research focused on the effects of stevia solely on biofilms of two streptococcal species separately, which does not fully replicate the complexity of the microbial community found in dental plaque. Therefore, further studies are needed to test the effect of stevia on other microorganisms and multi-microbial biofilms associated with caries such as *Streptococcus sobrinus* and different *Lactobacillus* and *Actinomyces* species.

With the findings of our study, we conclude that stevia has shown antibacterial potential by reducing biofilm formation, EPS production, and regulating the expression of biofilm-related genes (*gbpB* and *gtfB*) in streptococcal species. These effects may contribute to its potential therapeutic applications in controlling dental plaques and caries. However, further research is needed to understand fully the underlying mechanisms and to optimize the use of stevia in dental health.

## Figures and Tables

**Figure 1 dentistry-11-00267-f001:**
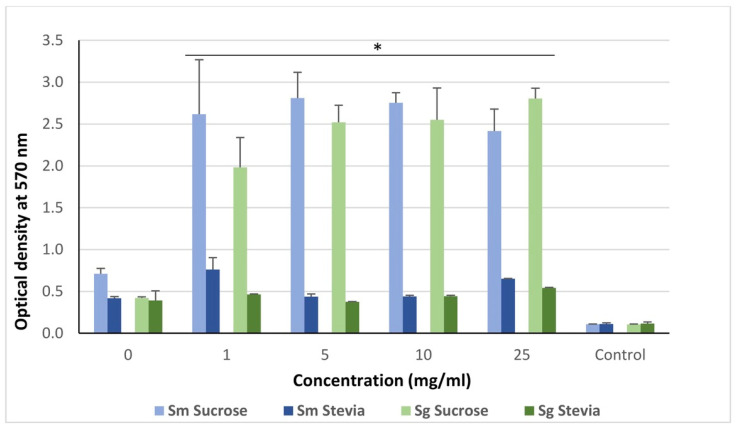
Crystal violet staining-based quantification of monospecies biofilms of *S. mutans* (Sm) and *S. gordonii* (Sg) treated with various concentrations of sucrose and stevia. Here, the control is culture medium (brucella broth) without Sm and Sg. The groups were compared by using Mann–Whitney U test. * *p* < 0.05.

**Figure 2 dentistry-11-00267-f002:**
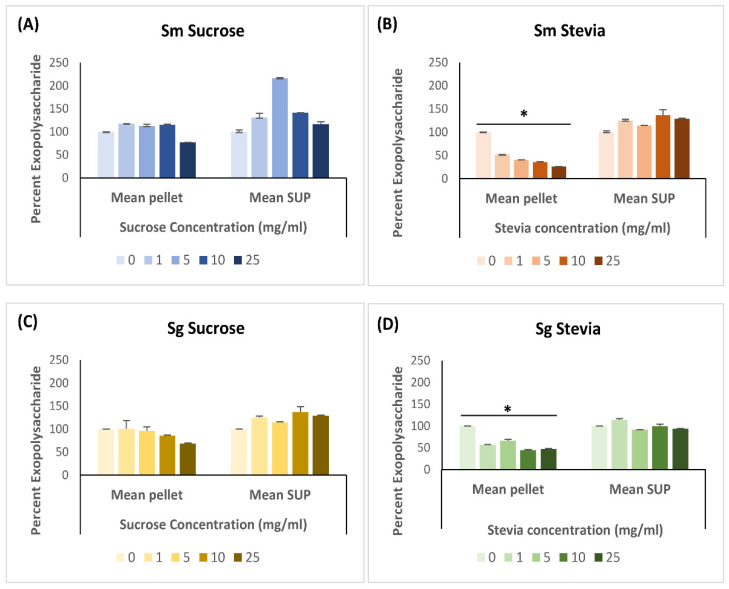
The effect of sucrose and stevia on exopolysaccharide production from *S. mutans* (Sm) (**A**,**B**); and *S. gordonii* (Sg) (**C**,**D**) biofilms. Total carbohydrate content in biofilms and biofilm supernatants (SUP) were quantified by phenol–sulfuric acid method. The data were analyzed by Mann Whitney U test. * *p* < 0.05.

**Figure 3 dentistry-11-00267-f003:**
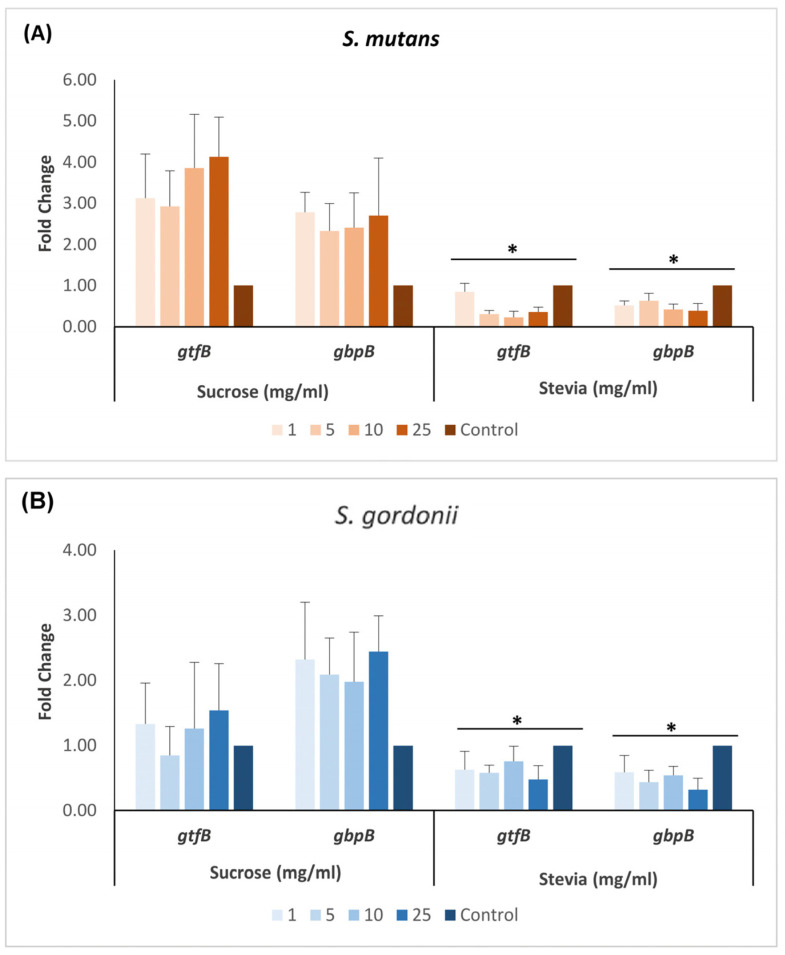
mRNA expression levels of *gtfB* and *gbpB* genes in sucrose- and stevia-treated biofilms of *S. mutans* (**A**) and *S. gordonii* (**B**). The data were analyzed by one-way ANOVA with Tukey post-hoc test. * *p* < 0.05.

## Data Availability

All relevant data are within the manuscript.

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
