# Peer review of "Sugar Substitute Stevia Inhibits Biofilm Formation, Exopolysaccharide Production, and Downregulates the Expression of Streptococcal Genes Involved in Exopolysaccharide Synthesis"

_dentistry, 2023, doi:10.3390/dj11120267_

Round 1

Reviewer 1 Report

Comments and Suggestions for Authors

This study investigates the effect of the sugar alternative Stevia on Streptococcus mutans and Streptococcus gordonii biofilm formation, EPS production, genes expression (gtfB and gbpB genes), which are essential for the synthesis of extracellular polysaccharides. The results showed that stevia can inhibit biofilm formation, reduce EPS production, and downregulate the expression of gtfB and gbpB genes in S. mutans and S. gordonii. It has certain clinical guiding significance, but the article need to improve.

1.      In method part, “Both Stevia and sucrose were added to the culture medium at different concentrations (0, 1, 5, 10 and 25 mg/ml) in two separate sets.” Why selection these concentrations? Besides, the concentrations of stevia and sucrose in figure 1 were 1, 2.5, 5 and 10 mg/ml ?

2.      In figure 1, “*P<0.05”, but no * found in the figure? The same problem existed in figure 2, please check.

3.      A P value of <0.05 was considered statistically significant

4.      Gene sequence used for PCR detection should list, and the internal reference genen of S. mutans and S. gordonii also need to describe in the method part.

Comments on the Quality of English Language

English language is fine, required minor editing.

Reviewer 2 Report

Comments and Suggestions for Authors

Dear Authors,

Well done on the good work. The topic is interesting and practical. The study is informative, well-presented and well illustrated. The abstract is concise, the materials and methods section is well-explained with excellent statistics. I would recommend adding more references to the discussion section. The conclusion is precise.

Best regards

Reviewer 3 Report

Comments and Suggestions for Authors

The paper intiltled "Impact of the Sugar Substitute Stevia on Biofilm Formation, Exopolysaccharide Production, and the Expression of Streptococcal Genes Involved in Exopolysaccharide Synthesis" contributes to the knowledge about biofilm formation during the expostion of Streptococcal species, Streptococcus mutans and Streptococcus gordonii, to different concentrations of sucrose and Stevia. The authors explore and evaluate several aspects of well-stablished biofilm-markers such as the exopolysaccharides (EPS) prodution and the expression of genes involved in EPS systhesis. The paper has an original contribution besides the lack of innovation in the scietific question/hipothesis. The results discuss the main proposed topic but not cleary shows the statistical correlations and discuss other secondaty aspects of the findings. The discussion is well-referenced and show correlation between the authors results and the current scientific literature, but need to be revised due an excess of uncorrelated information. Additionaly,the manuscript needs deep review and improvement in the English language. Based on the exposed, the manuscript shows a lot of promise, but some major issues need to be addressed before it can be published.

Title, lines 2-4: "Impact" is a neutral word that don't demonstrate any significance about the findings. Additionally, fhe full description of the assays performed in the title is not necessary. I suggest rework the title to show more direct and clear about the study findings.

Abtract:

Line 20: Please, adjust Streptococcus to Italic.

Key-words- line 33: to increase the discoverability of your paper use this space to put words different from the ones available in the title. If necessary you can check some recommeded keywords from existent lists.

Introduction:

In general, the introduction as presentend is repetitive (please, see comment in discussion for lines 209 - 237) and lacks cohesion. E.g., this is clearlly noted in the lines 51-58 where the sentences lack cohesive devices. Please, re-write the introduction to improve cohesion.

Line 36: Recently dental caries were reclassified from infections disease to a non-communicable chronic disease. Please, ajust the term. For your information some references:

- Pitts, N., Twetman, S., Fisher, J. et al. Understanding dental caries as a non-communicable disease. Br Dent J 231, 749–753 (2021). https://doi.org/10.1038/s41415-021-3775-4

- Giacaman et al., Understanding dental caries as a non-communicable and behavioral disease: Management implications. Frontiers in Oral Health.2022. DOI=10.3389/froh.2022.764479  

Lines 43-44: 'As besides' is not a common used English term. I suggest to adjust to or similar to: "In addition to its ability to be fermented by oral bacteria [..]"

Line 48: Please, add a reference for the following sentence: 'According to the ecological plaque hypothesis this low pH re-48 sults in an increase in cariogenic bacteria in the dental plaque'

Line 50: 'in situ' - Latim terms are usually written in Italic. Please, check the Journal guidelines for this.

Lines 66-67: Since you are mentioning the European Food Safety Authority, it´s expected direct mention to this institution. Samuel 2018 is a review, and could be used for the previous sentences but I suggest not use it as indirect reference. Please, adjust this reference for a direct citation from the European Food Safety Authority.

Lines 68-73: In the last paraghaph have two sentences about the genes gtfB and gtfC that are not conected to the previous paragraph context. Please, adjust it for improve cohesion.

Methods: 

Line 106: Please, describe or add a reference for the method used for the biofilm pH measurement.

Lines 109-110: In the sentence "The crystal 109 violet quantification results clearly indicated a notably greater biofilm biomass when sucrose was present." Please, add the results described in the Supplementary Material.

Subhead: 2.5 RNA extraction:

- Please, include the name of the kit used for this step mentioning also the manufacturer. Since the decribed protocol matchs with the manufacturer protocol is not necessary to describe the entire procedure, just mention that "the extraction of the RNA as performed accodingly to the manufacturer manual" or similar sentence. Only the last two sentences (Lines 129-131) on this subhead don't need to be changed. 

- Please, add a description for the extracted RNAstorage as well as its conditions.

Line 133: How long were the extracted RNA stored at -20ºC before proceed with the downstream procedures? For long storage samples, was the RNA tested for integrity?

Line 141: Please, clarify the correct anneling temperature for each pair of primer used.

Line 144 - 146: In regards to the Reference gene used, 16S rRNA, is there available a study or a validation data performed by the authours that tested the stability in the 16S rRNA expression when in different conditions, e.g., supplementation with Sucrose and/or Stevia? If yes, please, add in Methods section the information about the stability of the Reference gene 16S rRNA and the proper reference / validation information. If the testing for the stability of the Reference Gene was performed by the authors, please add this assay Method and Results in the Supplementary Material or add it to the main manuscript.

Line 146 to 148: Could I consider the technical and biological replications as described the same for all aforementioned tests?

Subhead: Statistics

- Did the authors submitted the data to the Normatility test (e.g., Shapiro-Wilk) before decide for the Mann Whitney U test or the one-way ANOVA for group comparison?

- The p letter is lowercase and italic. Please, correct it all manuscript.

Results

Lines 158 to 161: Where are the data for the pH analysis? If the authors prefer not add it in the main manuscript, please, add the pH results in the Supplemental Material. 

Fig. 1: Please, for this Figure, consider the coments below:

- Whats the difference between 0 (zero) sucrose or Stevia concentration and Control (columns in the right)? Is it the Negative Control? Please, include a description for the controls of this assay in the Methods.

- Please, include in the Figure 1 the unit used for the Sucrose and Stevia concentration. 

- Please, improve the description for the controls in the Figure 1 legend.

- Please, insert in the legend the statistic test used in the data set analyzed.

- Please, insert *(asterisk) in the data set which the statistic significance was observed.

- Please, change the species names to Italic.

- As a suggestion to make the Figure 1 graphic more clear, I suggest a bigger separation in the columns between the species.  The color set also can include similar colors for correlated species, e.g. light green for Sm sucrose and dark green for Sm Stevia. This suggestion is only to improve data intepretation. The authors can decide how to demonstrate their data set.

Lines 168 - 169: 

- In the result for S. mutans supernatant incubated with Stevia (Fig. 2) is showing a increased concentration for all concentrations (1 to 25 mg/mL) when compared to zero. This increase seems to be between 20% to 40%. Please, add the  data and statistical analysis for this data set also in table format. This new data can be added to the Supplementary Material. 

Lines 169-171: 

- By analyzing the Graphic, the higher Sucrose concentration (25 mg/mL) seems to have an inhibitory effect on biofilm mass production for both Streptococcus species. Based on the analysis of the bar sizes, seems to be a reduction between 20% and 30% which, if its proved, is significant. There´s a high change to this evaluation be proved in the statistic analysis because the Standard Deviation (SD) is tiny and means are different. Additionally, from the Statistical point of view, ANOVA test compare the means of groups and its post-test compare each samples against each other, thus, some difference is expected. PPlease, add the  data and statistical analysis for this data set also in table format. This new data can be added to the Supplementary Material. 

- Another interesting find which was unexplored in the Results was the increase (twice when compared to the 0 (zero) mg/mL) of exopolysaccharides in the supernant of S. mutans at 5 mg/mL sucrose concentration. This increase is followed by a decrease indicanting plato followed by inhibitory effects. Here we also have a small SD so a deeper statistical investigation is needed. I suggest to consider to highlight this result as well as its discussion on the proper section.

For figures 2 and 3: For Figures that contain more than one graphic, please, use the letter identification for each one and describe it properly in the legends. For Figure 3, e.g., the letter-code identification was mentioned in the legend but not implemented in the pictures.

Fig 2: Please, for this Figure, consider the coments below:

- Please, add the unit used for the Sucrose and Stevia concentration.

- In the Introduction and Methods, Stevia was written with the first letter in uppercase, in the legend of Figure is showed in lowercase. Please, consider to standardize the word "Stevia or stevia" in the manuscript.

- Add the abbreviations meaning in the figure legend.

- Please, insert in legend the statistic test used in the data set demonstrated. 

- Insert * (asterisk) in the date set which the statistic significance was observed.

- In the Graphic for Sm Stevia, the y axis Title is bigger than other pictures. Please, consider to standardize the captions.

Line 178: Please, put the gtfB in Italic.

Line 180: Please, put the gbpB in Italic.

Figure 3:  Please, for this Figure, consider the coments below:

- The resolution of the graphics is low. Please, upload higher quality images.

- Please, remove the -0.50 from the Y axis of the graphic and adjust the numbers to show the 1.

- Please, add the unit of the concentration for Sucrose and Stevia.

- The word 'fold' there´s no plural when describing changes in value or quantity. Thus, only "Fold Change" can be applied. However, a more scientific language can be applied by using "Level of Expression". Usually, the expression in the positive control, i.e., the expression of the bacteria grown in media with no addtion of Sucrose or Stevia, is considered as =1 and from this, the expression level is calculated for the supplemented-testing samples. Based on your graphic for S. gordonii, this procedure for level of expression calculation was followed, but the choice of numbering in the y axis for S. mutans don't allow to see it clearly. Please, amend it accordingly.

- In the S. mutans graphic, please, adjust the y axis size to avoid the extra space in the top (6-8-fold).

- A suggestion for better vizualiation of the graphic, please consider reorganize the columns to show the data grouped by gene expressed and not by the Sucrose x Stevia. This will allow the reader to better compare the expression data. However, the author can choose the best way to demonstrate their on data.

- The expression evaluated in the Sucrose treated group have a big SD. For this data, only two experiments in duplicate each was performed in the qPCR? Considering this level of variation, I suggest additional experiments (5 biological repetitions in triplicate each) to confirm the expression profile. The addition of a second Reference Gene is also recommended.

Discussion:

Line 195: Correct the word 'avid' to 'avoid'.

Lines 200-201: Please, improve description for the "The results", which results are the authors refering to? All? A specific data set?

Lines 201-204: Include the discussion for the increase in the EPS production in Sucrose treated groups (after confirmation by statistical analysis).

Line 204: In this sentence Streptococcal species is with uppercase, but in other mentions on the manuscript is in lowercase, please, standardize it.

Lines 204-205: Important to mention about the effect of the Sucrose in the expression of the genes.

Line 211: Please, remove the 'this' before extracellular matrix. 

Line 212: The first mention of the EPS abbreviation was done in the second paragraph of the discussion (line 197). Please, adjust the first mention for the proper place.

Line 209 - 237: These paragraphs are too long and not direct discuss the results alread presented or contribute for further disscuson. The presented information could contribute for improvement of the Introduction. Please, sintetize and keep only the information that is possible to correlate with the results presented in the manuscript.

Line 241: The use of temporal markers as "A very recent study" should be avoided. Please, consider rewrite it.

Line 243: Fist mention of a species needs to be the full-name, please, substitute C. albicans for Candida albicans (in italic).

Line 244: 'Also' is not formally used in the end of a sentence, is more used in informal or spoken English, please, rewrite to improve clarity and formal English language use.

Line 255: the expression "so on", follows the same understanding for previous explanation (Line 244), please, adjust it.

Line 267: Please, adjust S. rebaudiana to italic. If is the first mention, consider to add the full name of the species.

Lines 283 - 291 + 315: Please, adjust genes and species names to Italic.

Citations: The Journal Guidelines instruct the citations in the manuscrip as: "In the text, reference numbers should be placed in square brackets [ ], and placed before the punctuation; for example [1], [1–3] or [1,3]. For embedded citations in the text with pagination, use both parentheses and brackets to indicate the reference number and page numbers; for example [5] (p. 10). or [6] (pp. 101–105). Please, correct as recommended.

References: Lack a style in the references, including the type of letter used. Please, check the Journal Guidelines to correct it. 

Comments on the Quality of English Language

I strongly recommend the manuscript submittion to a certified English review to be presented in the next resubmission.

Reviewer 4 Report

Comments and Suggestions for Authors

This paper evaluated the effect of Stevia in the biofilm production and gene expression on S. mutans and S. gordonii. This topic has been investigated, and it is well known that Stevia is not fermentable and not metabolized by cariogenic bacteria. The methods were good, and the overall paper is well written. However, the manuscript could be improved if additional combination of bacterial species (Lactobacillus sp., Bifidobacterium sp., etc) and fungi, especially Candida albicans, were also evaluated, in single and, at least, dual species experiments. In addition, other sweeteners could have been included and compared. This would add value and novelty to the manuscript.

Methods:

Lns 85-90 are repeated in other sessions. Please remove it.

Ln 106: “The pH of biofilm was also measured.” How? Describe the method.

What was the formulation of the negative control (culture medium without sucrose a=or stevia)? Because it has some source of glucose to enable GTF to produce EPS. So, can this be considered a negative control?

I suggest the authors to read and include this reference - Augustinho do Nascimento C, Kim RR, Ferrari CR, de Souza BM, Braga AS, Magalhães AC. Effect of sweetener containing Stevia on the development of dental caries in enamel and dentin under a microcosm biofilm model. J Dent. 2021 Dec;115:103835. doi: 10.1016/j.jdent.2021.103835. Epub 2021 Oct 12. PMID: 34653536.

Round 2

Reviewer 3 Report

Comments and Suggestions for Authors

There's no additional comments or suggestions. The authors adjusted and improved the paper as suggested.

Author Response

Respected reviewer,

Thanks so much for your precious time on reviewing our manuscript. We believe that your strict scientific assessment has improved the paper. Thanks once again!

Best regards

M Karched

Reviewer 4 Report

Comments and Suggestions for Authors

N/A

Author Response

(The authors gave the same response as above.)
